# Change in the Electronic Structure of the Cobalt(II) Ion in a One-Dimensional Polymer with Flexible Linkers Induced by a Structural Phase Transition [note 1]

**DOI:** 10.3390/ijms24010215

**Published:** 2022-12-22

**Authors:** Dmitriy S. Yambulatov, Julia K. Voronina, Alexander S. Goloveshkin, Roman D. Svetogorov, Sergey L. Veber, Nikolay N. Efimov, Anna K. Matyukhina, Stanislav A. Nikolaevskii, Igor L. Eremenko, Mikhail A. Kiskin

**Affiliations:** 1N. S. Kurnakov Institute of General and Inorganic Chemistry, Russian Academy of Sciences, 31 Leninsky prosp., 119991 Moscow, Russia; 2A. N. Nesmeyanov Institute of Organoelement Compounds, Russian Academy of Sciences, 119991 Moscow, Russia; 3National Research Center “Kurchatov Institute”, 123182 Moscow, Russia; 4Institute “International Tomography Center”, Siberian Branch of Russian Academy of Sciences, 630090 Novosibirsk, Russia

**Keywords:** cobalt(II) complex, structural phase transition, orbital angular momentum, slow magnetic relaxation, dynamic crystals

## Abstract

A new 1D-coordination polymer [Co(Piv)_2_(NH_2_(CH_2_)_6_NH_2_)]_n_ (**1**, Piv is Me_3_CCO_2_^−^ anion) was obtained, the mononuclear fragments {Co(O_2_CR)_2_} within which are linked by μ-bridged molecules of hexamethylenediamine (NH_2_(CH_2_)_6_NH_2_). For this compound, two different monoclinic *C*2/*c* (α-**1**) and *P*2/*n* (β-**1**) phases were found at room temperature by single-crystal X-ray diffraction analysis, with a similar structure of chains and their packages in unit cells. The low-temperature phase (γ-**1**) of crystal **1** at 150 K corresponds to the triclinic space group *P*-1. As the temperature decreases, the structural phase transition (SPT) in the α-**1** and β-**1** crystals is accompanied by an increase in the crystal packing density caused by the rearrangements of both H-bonds and the nearest ligand environment of the cobalt atom (“octahedral CoN_2_O_4_ around the metal center at room temperature” → “pseudo-tetrahedral CoN_2_O_2_ at 150 K”). The SPT was confirmed by DSC in the temperature range 210–150 K; when heated above 220 K, anomalies in the behavior of the heat flow are observed, which may be associated with the reversibility of SPT; endo effects are observed up to 300 K. The SPT starts below 200 K. At 100 K, a mixture of phases was found in sample **1**: 27% α-**1** phase, 61% γ-**1** phase. In addition, at 100 K, 12% of the new δ-**1** phase was detected, which was identified from the diffraction pattern at 260 K upon subsequent heating: the *a*,*b*,*c*-parameters and unit cell volume are close to the structure parameters of γ-**1**, and the values of the α,β,γ-angles are significantly different. Further heating leads to a phase transition from δ-**1** to α-**1**, which both coexist at room temperature. According to the DC magnetometry data, during cooling and heating, the χ_M_*T*(*T*) curves for **1** form a hysteresis loop with ~110 K, in which the difference in the χ_M_*T* values reaches 9%. Ab initio calculations of the electronic structure of cobalt(II) in α-**1** and γ-**1** have been performed. Based on the EPR data at 10 K and the ab initio calculations, the behavior of the χ_M_*T*(*T*) curve for **1** was simulated in the temperature range of 2–150 K. It was found that **1** exhibits slow magnetic relaxation in a field of 1000 Oe.

## 1. Introduction

Materials based on molecular complexes, for which the bistability of the physical properties is realized, attract interest from a fundamental and practical point of view, as these properties can be used in the development of smart materials for storage, the processing and transmitting of data, as well as in sensors, electronic components, etc. [1,2,3]. Particular attention has been given to systems in which magnetic effects are realized, associated with changes in the spin state (spin crossover phenomenon, charge transfer induced spin transition, super-exchange), the presence of residual magnetization, magnetization hysteresis, slow magnetic relaxation, etc. [4,5,6,7,8,9,10]. Magnetic effects in crystals of complexes are due to the electronic structure of the metal ion, the coordination environment and the nature of the ligands, packing effects, and intra- and inter-molecular interactions.

The design of molecular complexes with 3*d* metal ions in the 3*d*^n^ (n = 4–7) configuration in combination with high-field ligands leads to compounds exhibiting spin-spin transitions between low spin (LS) and high spin (HS) states [11]. Often, valence tautomerism is observed for paramagnetic metal complexes with redox-active ligands [12]. As for molecular magnets, the key method is the control of the geometric parameters of the coordination environment of the metal centers in 4*f* and 3*d* metal complexes with orbital angular momentum. This approach should certainly be considered during the chemical assembly of single molecule (SMMs) and single ion (SIMs) magnets [13,14,15].

In addition, intermolecular interactions in crystals play a special role in determining a variety of the physicochemical characteristics of compounds, in particular, magnetism. For example, modulation of the spin state can be caused by co-crystallization involving halogen bonds [16,17], the appearance of interligand π-stacking [18,19], as well as the presence of hydrogen bonds between the ligands [20,21]. In some cases, these spin transitions are accompanied by structural phase transitions (SPTs), which are generally reversible, and are accompanied by a thermal hysteresis loop.

SPTs can occur both with minor changes in the coordination sphere of the metal ion and with the breaking of the M-L bond between the metal atom and the donor ligand atom. Using the example of cobalt(II) complexes with orbital angular momentum, it was shown that slight distortions of the coordination sphere of the metal atom are accompanied by changes in the magnetic behavior (the difference in χ_M_*T* values by ~3.5% in the temperature range 240–228 K with a hysteresis loop >10 K) [22]. However, a change in the geometry of the polyhedron as a result of breaking the M–L bond can lead to an increase in the energy gap between the excited and ground states, due to which χ_M_*T* forms a hysteresis loop of 14 K and the difference in χ_M_*T* in the range of 135–90 K is ~10.5% [23].

This paper describes a method for the synthesis of a new 1D polymer [Co(Piv)_2_(NH_2_(CH_2_)_6_NH_2_)]_n_ (**1**), in which the mononuclear fragments {Co(O_2_CR)_2_} are linked by μ-bridged molecules of hexamethylenediamine (NH_2_(CH_2_)_6_NH_2_). For **1**, possible crystal forms were studied at different temperatures. The magnetic behavior of polymer **1** was studied and it was shown that a decrease in temperature leads to SPT due to changes in the intra- and inter-molecular H-bonds and in the coordination number of the metal atom, as a result of which the energy of the singly occupied *d*-orbital of cobalt(II) decreases.

## 2. Results and Discussion

### 2.1. Synthesis and Characterization

The interaction of equimolar amounts of [Co(Piv)_2_]_n_ and hexamethylenediamine (NH_2_(CH_2_)_6_NH_2_)) in dry acetonitrile in an inert atmosphere led to the formation of violet crystals with a quantitative yield of the product with composition [Co(Piv)_2_(NH_2_(CH_2_)_6_NH_2_)]_n_ (**1**).

The obtained IR-spectra of **1** (Appendix A) containing stretching vibrations at 3316, 3234, 3160 cm^−1^ (ν(NH)), 2975 cm^−1^ (ν_as_(CH_3_/CH_2_), 2925 cm^−1^ (ν_as_(CH_3_/CH_2_), 2860 cm^−1^ (ν_s_(CH_3_/CH_2_), 1580 cm^−1^ (ν_as_(COO)), 1540 (ν_as_(COO)), 1470 cm^−1^ with shoulder at 1460 cm^−1^ (σ_as_(CH_3_) and/or σ(CH_2_)), 1410 cm^−1^ (ν_s_(COO)), 1360 cm^−1^ (σ_s_(CH_3_) and σ(C(CH_3_)_3_)), 1145 cm^−1^ (σ(C(CH_3_)_3_)), 1083 cm^−1^ (C–N stretching frequency of primary amines), 807 and 792 cm^−1^ (NH_2_ bending frequencies of primary amines) and 725 cm^−1^ (σ(CH_2_)) show the vibrations of aliphatic amine and trimethylacetate ligands [24,25,26]. We also compared the IR spectra of the solid sample isolated during the synthesis with the spectra of this sample at room temperature after one-month exposure to air (Appendix A) and found no significant changes. Thus, we can conclude that compound **1** is resistant to moisture and atmospheric oxygen.

### 2.2. Diffraction and DSC Studies of Polycrystals and Single Crystals

It turned out that the isolated crystals **1** are not crystallographically homogeneous at different temperatures. For crystals of this compound, three identified forms were found: two monoclinic (*C*2/*c*, α-**1**) and (*P*2/*n*, β-**1**) at room temperature (selected from a set of crystals obtained in one synthesis) and triclinic (*P*-1, γ-**1**) at 150 K (Table 1). Separately, it should be noted that the cooling of the α-**1** and β-**1** crystals at a rate of 5 K/min leads to their destruction; therefore, cooling was carried out at a rate of ~2 K/min, which made it possible to establish the structure of the γ-**1** phase, which is the same for both initial monoclinic forms at a low temperature.

The DSC data in the temperature range 273–140 K (cooling process) for **1** showed two anomalies (exo-effects), below 250 and 205 K. The second one takes place in the temperature range 205–150 K. During heating, the anomalies with endo-effects start above 225 K and continue up to 300 K (Figure 1). It can be assumed that the observed structural changes (see below) from α-**1** to γ-**1** begin below 205 K (exo-effect in the region of 205–150 K), and the reverse phase transition(s) is(are) shifted to the region above 225 K. To attribute the anomalies observed in the DSC, variable-temperature powder X-ray diffraction studies (VT-PXRD) were carried out on a polycrystalline sample.

The simulation of the powder diffraction pattern taken at room temperature showed that all of the observed peaks correspond to the structure with the space group *C*2/*c* (α-**1** phase) (Appendix A), while the description of the experimental data, only by the second structural model with the *P*2/*n* group (β-**1** phase), is markedly worse (Appendix A). Based on this, we can conclude that the crystalline sample obtained in the synthesis is predominantly a rather pure α-**1** phase; therefore, in order to understand the properties of sample **1**, we further focused on the prevailing presence of the α-**1** phase. Compound **1** is stable at RT and maintains its crystallinity for a long time (more than 3 months).

VT-PXRD for **1** was applied in the temperature ranges 300–100 K (cooling) and 100–300 K (heating) (Figure 2 and Appendix A). When the sample is cooled with a step of 40 K, the XRD patterns change below 200 K. In this case, the peaks related to the γ-**1** phase appear; however, in addition to them, a number of peaks of the new δ-**1** phase are observed on the diffraction patterns. The combination of the data sets shows that the concentration of γ-**1** increases with the decrease in temperature (35%, 58%, 61% at 180, 140 and 100 K, respectively; Figure 2, Appendix A; Table 2), the content of the initial phase α-**1** decreases (54%, 28%, 27% at 180, 140 and 100 K, respectively; Figure 2, Appendix A; Table 2), and the intensity of the δ-**1** peaks does not change significantly with further cooling (the content of δ-**1** is in the range of 11–14%). When heated above 180 K, the intensity of the peaks of the δ-**1** phase increases with the simultaneous disappearance of the γ-**1** peaks. The unit cell parameters of the δ-**1** phase were determined from the XRD data at 260 K (Figure 2 and Appendix A). Despite the close values of the *a*,*b*,*c*-parameters and the unit cell volumes of the γ-**1** and δ-**1** phases, they differ in α,β,γ-angles, which indicates the uniqueness of the δ-**1** phase (Table 1). The maximum amount of δ-**1** was recorded at 220–260 K (77% δ-**1**, 23% α-**1**). Subsequent heating to 300 K resulted in the formation of α-**1** with δ-**1** impurity (19% δ-**1**, 81% α-**1**; Figure 2 and Appendix A). Presumably, the δ-**1** → α-**1** transition is accompanied by endo-effects on the DSC curve above 270 K (Figure 1, heating).

It was shown that the cooling and holding of sample **1** at temperatures below the onset of the phase transition (77 and ~180 K), followed by heating to RT, leads to the formation of a mixture of α-**1** and δ-**1** phases in the ratio δ-**1**:α-**1** = 66:34 at 77 K and δ-**1**:α-**1** = 43:57 at 180 K (Appendix A). The dependence of the δ-**1** concentration on the cooling temperature suggests that this phase is formed as a result of the SPT of the γ-**1** phase. Thus, it is likely that the δ-**1** phase is an SPT intermediate (heating) of γ-**1** into α-**1**.

It was subsequently discovered that crystals **1** (α-**1** and β-**1** phases) contain molecules of a coordination linear polymer in which the hexamethylenediamine bridges link the cobalt atoms, each of which is linked to two carboxylic acid ligands (Figure 3a and Appendix A). It is important to note that, in these phases, the Co(II) atom has an octahedral ligand environment (CoN_2_O_4_ chromophore; α-**1** structure contains two independent Co atoms with a close environment; Appendix A). The Co–O distances with the chelate carboxylate group (in α-**1** (Co–O 2.085(4)‒2.260(4) Å) and β-**1** (Co‒O 2.072(17)‒2.278(11) Å) differ somewhat from each other in value (~0.2 Å), which is often observed in cobalt(II) carboxylate complexes (see for example [27,28,29,30,31]). As the temperature decreases (150 K), the octahedral environment of the metal center in the γ-**1** structure rearranges into a distorted trigonal-bipyramidal environment (CoN_2_O_3_ chromophore, Figure 3b, Appendix A) as a result of the formal breaking of the Co‒O bond between the metal ion and the oxygen atom of one of the chelate pivalate ligands, due to the rotation of the chelate carboxyl group (Co…O 2.622 Å). However, the process does not stop there: in the low-temperature structure γ-**1**, a noticeable elongation of the Co-O bond from the second chelate carboxyl group (Co…O 2.369 Å) is observed, which allows us to interpret the environment (certainly with a certain degree of stipulation) of the Co(II) metal centers as distorted tetrahedral (CoN_2_O_2_ chromophore, Appendix A). At the same time, the Co-N bond lengths change insignificantly (2.098(4) and 2.099(4) Å for α-**1**, 2.087(6) Å for β-**1**, 2.029(11) and 2.088(10) Å for γ-**1**).

The geometry of the chains in the crystals of the high-temperature α-**1** and β-**1** phases corresponds to a zigzag pattern (the angle between cobalt atoms is 148.2° for both structures), while in the low-temperature γ-**1** phase the chain is close to straight (the corresponding angle is 169.8°). This is due to a change in the metal polyhedron and a change in the conformation of the bridging ligand NH_2_(CH_2_)_6_NH_2_. In α-**1** and β-**1**, the metal atom lies on the crystallographic axis 2, the substituent chains are symmetrical, the Co-N-C-C torsion angle is 178.8(3)° (for Co1) and 170.3(3)° (for Co2) for α-**1** and 174.2(5)° for β-**1**. As a result, the structure of the polymer chains in the α-**1** crystal differs slightly from that in β-**1** (Figure 4a). As the temperature decreases, the symmetry of the crystal decreases, the chain conformations on the opposite sides of the metal also change and become different, and the Co1-N-C-C angles are 170.2(9) and 77.6(3)° in the high-temperature and low-temperature phases, respectively. As a consequence, the Co…Co distances within the chain change; at room temperature, they are 12.432(2) Å in α-**1** and 12.431(2) Å in β-**1**, but 12.590(6) Å and 11.002(6) Å in **γ**-**1**. As a result, the geometry of the polymer chain at different temperatures differs significantly (Figure 4b).

The packing of the chains in all cases represents parallel infinite layers formed by a NH…O type of hydrogen bonds (Table 3). In the crystals at RT and 150 K, each pair of coordinated amino groups form two H-bonds with the oxygen atoms of the carboxylate groups of the adjacent chain, forming supramolecular layers (Figure 3). Additionally, in γ-**1**, the oxygen atom of the carboxylate group forms an intramolecular H-bond with the hydrogen atom of the NH_2_ group of the bridging ligand (Figure 3b). The comparison of the intermolecular H-bonds upon transition from α-**1** to γ-**1** revealed a shortening of the N1…O interatomic distance by 0.03 Å and an elongation of N2…O by 0.07 Å (Figure 3, Table 3). Structural changes in the layer are accompanied by a decrease in the minimum Co…Co distance (by 0.1–0.2 Å) between atoms of neighboring chains: 5.710(2) Å in α-**1**, 5.622(2) Å in β-**1**, and 5.510(6) Å in **γ**-**1**.

The supramolecular layers are linked to each other by van der Waal’s interactions (Appendix A). The distances between the planes drawn through the metal atoms in the layer are close and amount to 8.041(2) Å in α-**1**, 8.050(2) Å in β-**1**, and 8.102(6) Å in γ-**1**. Thus, the changes associated with an increase in the density of the crystals upon the decrease in the temperature are determined by the intralayer compression of the chains.

### 2.3. Spectral Properties

The IR spectra of the samples maintained at 77 and ~180 K are identical (Appendix A).

Changes are observed in the diffuse reflectance spectra at RT (Appendix A). Sample **1** has a broad band (400–800 nm) with a maximum at 534 nm and shoulders at 480, 500, and 680 nm. For sample **1**_77K_ (sample **1** after exposure at 77 K for 5 h), a shift of the maximum of a wide band (538 nm) is observed, with shoulders at 480, 500, 575, and 690 nm. The spectral bands correspond to d-d transitions for a high-spin cobalt(II) ion surrounded by high-field ligands [32,33,34], which confirms the splitting of the characteristic band at ~500 nm. The differences in the spectra during the transition from **1** to **1**_77K_ indicate changes in the electronic structure of the cobalt(II), which may be associated with the structural changes described above; namely, the formation of a mixture of δ-**1** and α-**1** phases, in which the ligand environment of the cobalt(II) ions changes along with the formal composition retention.

### 2.4. EPR Spectroscopy, DC Magnetic Data and Theoretical Calculations

According to the X-band EPR spectroscopy at a temperature of 10 K, a signal was registered for **1** (Appendix A), which was simulated using the software package *EasySpin* [35]. The complex is characterized by typical values of the spin Hamiltonian of the high-spin Co^II^ ion. The spectrum simulation in the effective spin model *S* = 1/2 made it possible to determine the parameters for the Kramer’s doublet: *g*_x_ = *g*_y_ = 4.6, *g*_z_ = 2.11, linewidth 30 mT. For the effective spin model *S* = 3/2, the following parameters of the spin Hamiltonian were obtained as a result of the best agreement between the theoretical model and the experimental data: *g*_x_ = *g*_y_ = 2.29, *g*_z_ = 2.18 (*g*_iso_ = 2.25), linewidth 30 mT. The estimated value of the magnetic anisotropy parameter D of the cobalt(II) ion used in the simulation was ~23 cm^−1^ (the assumed value is much larger than the microwave quantum and cannot be determined from the X-band EPR spectrum). The parameter D has a positive sign, i.e., The Kramer’s doublet ±1/2 is the main one. The rhombicity parameter E of the splitting tensor in the zero magnetic field was set to zero.

The magnetic properties of **1** were studied in both static and dynamic modes. Measurements of the static magnetic susceptibility χ were carried out in a magnetic field with strength *H* = 5 kOe. The value of χ_M_T at a temperature of 300 K is 2.88 cm^3^K/mol (Figure 5a), which significantly exceeds the theoretical pure spin value for the Co^2+^ ion (^4^F_9/2_, *d*^7^, *C* = 1.90 cm^3^K/mol). It can be expected that such an overestimation of the χ_M_T values is due to the nonzero orbital contribution. As the temperature drops below 190 K, a noticeable change in the value of χ_M_T (Figure 5a), from 2.84 cm^3^K/mol (at 192 K) to 2.61 cm^3^K/mol (at 162 K), is observed, which, as has already been shown by the single crystal and powder X-ray diffraction studies and calculations, is due to a change in the packing of the polymer chains upon the change in temperature with a corresponding change in the geometric parameters (bonds and angles) in the coordination sphere around the cobalt ions, as well as in the *g*-tensor and the zero-field tensor (parameters D and E). According to the DC magnetometry data, the value of χ_M_T for **1** sharply decreases by 8% in the range of 200–150 K, which corresponds to the SPT and to the change in the magnetic behavior of the cobalt(II) ions upon the rearrangement of the coordination environment of the metal atom.

χ_M_T(T) curve calculations were performed for high- (192–300 K) and low-temperature (2–146 K) regions, taking into account the zero-field splitting of Co^2+^ electronic levels, in the Mjolnir [36,37] and PHI programs [38], respectively.

The high temperature region is described using the Griffith Hamiltonian (GH) [39,40,41,42], which takes into account the contribution of unquenched orbital angular momentum for octahedral Co(II) complexes (1):(1)$$\hat{H}=-\frac{3}{2}\kappa \cdot \lambda \left(\vec{S}\cdot \vec{L}\right)+\Delta_{\mathrm{ax}}\left({\hat{L}}_{z}^{2}-\frac{1}{3}L\left(L+1\right)\right)+\Delta_{\mathrm{rh}}\left({\hat{L}}_{x}^{2}-{\hat{L}}_{y}^{2}\right)+\mu_{B}\vec{B}\left(g_{e}\vec{S}-\frac{3}{2}\kappa \vec{L}\right)$$
where Δ_ax_ and Δ_rh_ = splitting of the ^4^T_1g_ ground term coming from symmetry lowering, κ = orbital reduction factor (it can vary between 0.6 and 1.0 depending on the complex), λ = spin-orbit coupling, *L* and *S* are the orbital angular momentum and spin operators, *g*_e_ = free-electron *g*-factor (2.0023), μ_B_ is the Bohr magneton, and *B* is the applied magnetic field. The final plot of χ_M_*T*(*T*) was simulated, taking into account the average parameters Δ_ax_ = −1316.9 cm^−1^, Δ*_rh_* = −153.7 cm^−1^ and λ = −174.54 cm^−1^ taken form the ab initio calculations (see details below, Table 4) and κ = 0.8 (Figure 5a). In addition, these values are quite consistent with those calculated for the distorted octahedral cobalt(II) complexes with “easy axis” anisotropy [43,44,45,46,47,48,49,50].

For the low-temperature region, the course of the curve is described by the parameters of the effective spin Hamiltonian, which takes into account the anisotropic parameter D (2):(2)$$\hat{H}=D{\hat{S}}^{2}+g_{\mathrm{iso}}\beta \hat{S}H$$

The calculated values were *g*_iso_ = 2.322(1), D = 14.3 cm^−1^, *R*^2^ = 3.9∙10^−5^ (Figure 5a). The value of axial parameter D is typical for Co^2+^ complexes, where metal ions are in a tetrahedral coordination environment [51,52,53,54].

To fully explain the *dc*-results, the parameters of the GH (1) and the effective spin Hamiltonian (3) were calculated using the CASSCF/NEVPT2 methods in the Orca 5.0.1 program [55]:(3)$$\hat{H}=g\mu_{B}BS+D\left[{\hat{S}}_{z}^{2}-\frac{1}{3}S\left(S+1\right)\right]+E\left({\hat{S}}_{x}^{2}-{\hat{S}}_{y}^{2}\right)$$
where *μ_B_—*Bohr’s magneton, *B—*applied magnetic field, *S—*full spin, D and E*—*axial and rhombic parameters of zero field splitting (ZFS), *Ŝ—*full spin operator.

In the transition from α-**1** to γ-**1,** an increase in the *z*-component of the *g*-tensor was observed (Table 4). The calculated SOC constants are of the same order as the constant of the free Co^2+^ ion (ζ = 533 cm^−1^). The calculated values of *g*_iso_ and D for γ-**1** are close to those calculated from the EPR data (*g*_iso_ = 2.25, D ≈ 23 cm^−1^) for the model with spin *S* = 3/2.

The comparison of the results of the calculations and the simulation of the magnetic data confirms that SPT is accompanied by the extinction of the orbital angular momentum and a decrease in SOC for Co^2+^ ions, the SOC constants correspond to 523.61 and 520.43 cm^−1^ for α-**1** and γ-**1**, respectively.

The structure of the high-temperature phase contains two crystallography non-equivalent cobalt atoms, each having a pseudo-octahedral environment with an axial elongation of Co-O bonds (Figure 3). In phase α-**1**, upon the application of a O_h_ crystal field (CF), the ground term ^4^F splits to ^4^T_1g_, ^4^T_2g_ and ^2^A_g_, where ^4^T_1g_ is the lowest in energy. As a result, lowering the symmetry of each Co^2+^ ion from ideal O_h_ to D_4h_ lifts the degeneration of the state ^4^T_1g_ by ^4^E_g_ and ^4^A_2g_ (Table 5), and due to the rhombic distortion, the ^4^E_g_ ground term is split in two [40,41,42,43,44,48,56].

In phase γ-**1**, upon the application of a T_d_ crystal field, the ground term ^4^F splits to ^4^A_2_, ^4^T_2_ and ^4^T_1_, where ^4^A_2_ is the lowest in energy. The perfect symmetry distortion from T_d_ to D_2d_ splits the ^4^T_2_ term into the ^4^B_2_ and ^4^E terms (Table 5). State ^4^A_2_ changes to ^4^B_1_ due to the CF symmetry [54,57,58,59]. In the case of γ-**1** complex, a significant increase in the ^4^B_2_ level compared to ^4^B_1_ induces easy-plane magnetic anisotropy.

In other words, lowering the temperature affects the splitting of the CF, due to a decrease in the orbital angular momentum and the SOC of the metal ions, the degeneracy of the excited states decreases.

As the electronic structure of both of the Co^2+^ ions in α-**1** are described by the GH, their spin-orbit contribution to the total magnetic moment, in this case, is considered as a superposition of the values for the ions.

To estimate the effect of the CF on the magnetic anisotropy, the relative energies of the d-orbitals were calculated using the AILFT method [60]. For α-**1**, the orbital energies were calculated for each Co^2+^ ion from the symmetry of CF (Figure 6). The three low-lying orbitals make up the partially occupied *t_2g_* subshell separated from the singly occupied *e_g_* subshell [47,61,62].

Lowering the temperature leads to the alignment of the polymer chains in the crystal, which leads to the same symmetry of the orbitals for γ-**1** (Figure 6). In the case of the Co1^2+^ ion, the breaking of the Co–O bond is accompanied by the removal of the carboxylate group along the y axis, which leads to the strong decrease in the energy of the *d*_yz_ and *d*_x2-y2_ orbitals. For the Co2^2+^ ion, the formation of a strong σ-bond with the second oxygen atom of the carboxylate group is observed. This probably affects the increase in the energy of the *d*_yz_ and *d*_x2–y2_ orbitals.

The doubly occupied *d*_xy_ and *d*_z2_ orbitals forming the *e* subshell are almost degenerate and stabilized with respect to the singly occupied high-lying *t_2_* (*d*_xz_, *d*_x2-y2_, *d*_yz_) orbitals [63,64,65].

As a reversible phase transition is observed for **1**, which is accompanied by a change in the electronic state of cobalt(II) ions observed in the behavior of χ_M_*T*(*T*), this effect can give rise to a thermal hysteresis loop in the magnetic susceptibility. To verify this assumption, we carried out additional measurements of the static magnetic susceptibility in a magnetic field of *H* = 1 kOe with a cooling of the sample from 300 to 2 K and subsequent heating up to 300 K. The behavior of the χ_M_*T*(*T*) curve upon cooling ((χ_M_*T*)_cool_) (Figure 7) is similar to the results described above (Figure 5a). The χ_M_*T*(*T*) curves during cooling and heating ((χ_M_*T*)_heat_) coincide in the temperature range of 2‒170 K. Above 170 K, the (χ_M_*T*)_heat_ values are lower than the (χ_M_*T*)_cool_ values: the difference between the (χ_M_*T*)_cool_ and (χ_M_*T*)_heat_ values in the temperature range 190–285 K is 8‒9%. The (χ_M_*T*)_heat_ curve up to 300 K does not reach the initial value (2.98 cm^3^K/mol vs. 3.13 cm^3^K/mol); as a result, an incomplete hysteresis loop of ~110 K is formed in the temperature range 170–300 K. These results are consistent with the incomplete phase transition at 300 K, as the magnetic moment for the superposition of the two phases, α-**1** and δ-**1,** is expected to be lower than for the α-**1** phase, provided that the electronic structure of the cobalt(II) ions in δ-**1** differs from α-**1**, which is confirmed by the difference in the diffuse reflectance spectra for pure α-**1** and a mixture of α-**1** and δ-**1** at RT (Appendix A).

### 2.5. AC Magnetic Data

In order to study the dynamics of the magnetic behavior of **1**, measurements of the dynamic magnetic susceptibility were carried out. The measurements in a zero external magnetic field showed the absence of significant values of the imaginary component of the dynamic magnetic susceptibility at *T* = 2 K, which indicates a rather high relaxation rate of the system, even at such a low temperature, and/or is a consequence of the fact that the splitting parameter in the zero field D has a positive sign and the ground state is the state with *m*_J_ = ±1/2. It is well known that the application of an external magnetic field can significantly increase the magnetic relaxation time due to the suppression of the effect of quantum tunneling of magnetization (QTM).

Measurements of the dynamic magnetic susceptibility in external magnetic fields that are different from zero make it possible to detect significant values in the frequency dependences of the imaginary component of the dynamic magnetic susceptibility at a temperature of 2 K, and, consequently, slow magnetic relaxation in complex **1** under these conditions (Appendix A).

Varying the strength of the external magnetic field made it possible to determine the optimal value at which the QTM effect is most effectively suppressed, however, relaxation by the direct mechanism is the least likely to occur. For complex **1**, the optimal value of *H* is 1000 Oe. The measurements of the ac magnetic susceptibility and approximation of the out-of-phase frequency dependences using the generalized Debye model (Figure 8) were used to plot the dependence of the relaxation time τ on *T* (Figure 9).

Approximation of the high-temperature linear part of the dependence τ(1/*T*) by the Arrhenius equation (τ = τ_0_·exp{Δ*E*/*kT*}) made it possible to estimate both the effective value of the remagnetization barrier and the fastest relaxation time in the system (τ_0_ = 6.4·10^−10^ s, Δ*E*/*k* = 30 K). The deviation of the dependence τ(1/*T*) in the semi-logarithmic coordinates from the linear one (Figure 9) points to the fact that, in addition to the above-barrier Orbach mechanism, relaxation passes through other possible paths. The best agreement between the theoretical curve and the experimental data was obtained for a linear combination of the Raman mechanism (τ_Raman_^−1^ = *C*_Raman_·*T^n^*^_Raman^) and direct (τ_direct_^−1^ = *A*_direct_*H^n^*^_direct^*T*) relaxation mechanism with the following parameter values: *C*_Raman_ = 1.9 K^-n_Raman^c^−1^ (±0.7), *n_*_Raman_= 9.5 (±0.4) and *A*_direct_= 2.4·10^−9^ K^−1^Oe^−n_direct^s^−1^ (±2·10^−11^), *n*__direct_= 4 (fixed for Kramer’s ion). The value of the parameter *n*_Raman_ obtained as a result of the best approximation is typical for the Kramer’s ions.

### 2.6. Discussions

For the resulting 1D polymer **1**, a reversible SPT is observed with a change in the coordination environment of the metal center, which is the Co(II) ion, from octahedral to tetrahedral in the temperature range 205–150 K (according to DSC data), this rearrangement is stabilized by the transition of the coordinated carboxylate group from the chelate mode to the monodentate mode and the formation of an intramolecular H-bond. SPT was confirmed by single-crystal and powder X-ray diffraction data at different temperatures. Upon cooling and subsequent heating, the crystal loses its crystallinity. According to XRD data, cooling of polycrystalline sample **1** below 200 K is accompanied by the formation of the γ-**1** phase. Cooling of **1** to 100 K leads to the formation of a mixture of phases γ-**1** (61%), established by the method of single-crystal X-ray diffraction analysis, δ-**1** (12%), and the initial α-**1** (27%). The presence of the δ-**1** phase in sample **1** at 100 K may be due to keeping the sample during the recording of the diffraction pattern at 180 K (the content of δ-**1** is ~12% at 180, 140, and 100 K). This agrees with the experimental results when sample **1** is heated in the range of 100–300 K: the phase ratio α-**1**:γ-**1**:δ-**1** at 100 and 140 K remains the same; above 140 K, the γ-**1** phase begins to transform into δ-**1**. The maximum concentration of δ-**1** (77%) in sample **1** was recorded at 220 and 260 K. Above 260 K, the δ-**1** phase transforms into α-**1** (270 K according to DSC data). The δ-**1** and α-**1** phases coexist at RT, what is confirmed by XRD data at RT for samples kept at 77 K (δ-**1**:α-**1** = 66:34) and 180 K (δ-**1**:α-**1** = 43:57). An increase in the concentration of δ-**1** in sample **1** kept at 77 K confirms the transition of γ-**1** to δ-**1**. Both phases are triclinic and have close *a*,*b*,*c* unit cell parameters and volume, but differ in the values of α,β,γ angles. Based on these data, the following scheme of polymorphic transformations was proposed (Figure 1):

The described phase transition, which leads to a change in the geometry of the coordination environment of the cobalt(II) ion in **1** (from octahedral for α-**1** to pseudo-tetrahedral for γ-**1**), significantly affects the magnetic behavior of sample **1** by an 8% drop in χ_M_*T* in the region of 200-150 K. According to the results of the ab initio calculations, SPT is accompanied by the extinction of the orbital angular momentum and a decrease in the SOC of the cobalt(II) ions, the SOC constants correspond to 523.61 and 520.43 cm^−1^ for α-**1** and γ-**1**, respectively. The difference in the electronic structure of the cobalt(II) ions in α-**1** and γ-**1** and the reverse phase transition being shifted higher in temperature through the δ-**1** phase leads to the formation of a hysteresis loop of ~110 K on the χ_M_*T* curve. The completion of the hysteresis loop is assumed to be above 300 K, as the presence of the δ-**1** phase (19%) in sample **1** at 300 K underestimates the value of χ_M_*T*. Furthermore, as the change in the values of (χ_M_*T*)_cool_ and (χ_M_*T*)_heat_ in the temperature range of 190–285 K is symbatic, and the cell parameters for γ-**1** and δ-**1** are close, we can assume that the coordination environment and the electronic structure of the cobalt(II) ions are similar in these phases.

The values of the *g*_iso_ (2.238) and D (17.555 cm^−1^) parameters for **1** at 150 K, obtained from the ab initio calculations, correspond to the EPR data at 10 K (*g*_iso_ = 2.25, D = ~23 cm^−1^) for the model with spin *S* = 3/2.

At helium temperatures, sample **1** exhibits slow magnetic relaxation in a field of 1000 Oe, which is often observed for cobalt(II) complexes with planar magnetic anisotropy (D > 0), for which spin-lattice relaxation occurs between lower levels *M*_s_ = ½ [66,67,68,69,70]. The observed magnetic relaxation in **1** is described by the sum of the direct mechanism and the Raman relaxation, which is consistent with the results of the ab initio calculations and the approximation of the DC data.

### 2.7. Conclusions

Thus, in this work, a new 1D polymer was synthesized and studied, in which the cobalt(II) atoms are linked into a chain by an aliphatic bridge, hexamethylenediamine. For this compound, a structural phase transition is observed below 205 K, which is accompanied by changes in the coordination environment of the cobalt(II) ions, the intra- and intermolecular H-bonds, and the packing density of the molecules in the crystal. We believe that the result of such structural rearrangements is a decrease in the value of χ_M_*T* by 8% in the region of 200–150 K, which is due to the extinction of the orbital angular momentum of the cobalt(II) ions. According to the results of the multi-temperature X-ray diffraction analysis (RT and 150 K), the VT-PXRD (100–300 K), and the DSC (135–300 K), it was shown that, for the initial monoclinic phases α-**1** and β-**1**, a SPT below 200 K is observed upon cooling, as a result of which the triclinic phase γ-**1** is formed, from which, when heated above 140 K, the triclinic phase δ-**1** is formed with close values of the unit cell parameters, and above 260 K, the initial monoclinic phase α-**1** is formed. The δ-**1** and α-**1** phases coexist at 300 K. The shift of the reverse structural phase transition is higher in temperature and the supposed similar electronic structure of the cobalt(II) ions in γ-**1** and δ-**1** causes the formation of a hysteresis loop of ~110 K on the χ_M_*T*(*T*) curve, in the range of 170–300 K, which is caused by reversible changes in the orbital angular momentum of the metal center as a result of the structural cooperative effects in the crystal and the dynamics of the Co–O bonds, similar to the previously described structural-magnetic transitions of cobalt(II) complexes [22,23]. Thus, perhaps this is the third example of the control of orbital angular momentum of cobalt(II) ions through a structural phase transition, which resulted in the formation of a hysteresis loop on the magnetic susceptibility with a record value of 110 K.

The analysis of the magnetic behavior of sample **1** in the temperature range of 150–2 K, EPR spectrum of **1** at 10 K, and ab initio calculations consistently led to a positive ZFS value for the cobalt(II) ion in the tetrahedral CoO_2_N_2_ environment. the easy-plane magnetic anisotropy for Co(II) in 1 at low temperatures is consistent with the observed slow magnetic relaxation, which is realized within the framework of a model that includes the sum of the Raman and direct relaxation mechanisms.

This study will continue with the aim of determining the completeness of the reversibility of the δ-**1** to α-**1** transition and the accompanying temperature changes in sample **1** over a wide temperature range. As part of the subsequent work, it is planned to refine the δ-**1** structure based on the XRD data and a set of studies with multi-temperature IR experiments, DC magnetic measurements, DSC and PXRD, affecting the high-temperature region >300 K.

## 3. Methods and Materials

### 3.1. Main Methods

The IR spectra of the compound was recorded on a Perkin Elmer Spectrum 65 spectrophotometer equipped with a Quest ATR Accessory (Specac) by the attenuated total reflectance (ATR) in the range of 400–4000 cm^−1^. Elemental analysis of the resulting compounds was carried out with a Carlo Erba automatic C,H,N,S-analyzer.

The EPR spectra were collected using a Bruker Elexsys E580 spectrometer at X-band (9 GHz) in continuous wave mode. The spectrometer was equipped with an Oxford Instruments temperature control system, and the powder polycrystalline spectra measured at *T* = 10 K.

Magnetic susceptibility measurements were performed with a Quantum Design susceptometer PPMS-9. The temperature dependences of the magnetization (*M*) were measured in 1000 and 5000 Oe magnetic fields in the temperature range 2–300 K during cooling, at a cooling rate of 1 K/min. During the AC susceptibility measurements in the frequency range of 10–10^5^ Hz, an alternating magnetic field amplitude was *H*_ac_ = 1–5 Oe. The measurements were carried out for the samples moistened with mineral oil to prevent any texturizing of the particles in DC magnetic field. The prepared samples were sealed in polyethylene bags. The paramagnetic components of the magnetic susceptibility χ were determined taking into account the diamagnetic contribution evaluated from Pascal’s constants as well as the contributions of the sample holder and mineral oil.

UV-visible diffuse reflectance spectra were recorded on a JASCO V-750 UV-VIS spectrometer with Horizontal Integrating sphere (PIV-756).

Differential scanning calorimetry (DSC) was performed on differential scanning calorimeter, DSC-60 Plus, at a cooling/heating rate of 10 K/min.

### 3.2. Materials and Synthesis

All manipulations were carried out under an inert atmosphere using vacuumed evacuated glass ampoules accompanied with an argon-filled glovebox. The acetonitrile was dried using the common method: over P_2_O_5_, kept on the molecular sieves (4 Å) and withdrawn by condensation just before the synthesis. The cobalt trimethylacetate [Co(Piv)_2_]_n_ was synthesized with a reaction of cobalt acetate tetrahydrate and an excess of pivalic acid. The Hexamethylenediamine (Aldrich, 98%) was purified by sublimation before use. The dry product **1** is stable to oxygen and air moisture.

#### Synthesis of [Co_2_(Piv)_2_(NH_2_(CH_2_)_6_NH_2_)]_n_ (**1**)

Weighed portions of [Co(Piv)_2_]_n_ (0.260 g, 1.000 mmol) and NH_2_C_6_H_12_NH_2_ (0.116 g, 1.000 mmol) were placed in a glass ampoule and degassed in a dynamic vacuum for 5 min (ampoule was cooled with liquid nitrogen to prevent diamine evaporation), the acetonitrile (15 mL) was condensed into the ampoule, the latter was fire-sealed and heated in an oil bath (120 °C) for six hours. The color of the reaction mixture changed from violet (initial Co(II) trimethylacetate) to colorless with a formation of block pink crystals precipitate (0.359 g, 95%). Anal. calculated for C_16_H_34_CoN_2_O_4_ (377.38) C, 50.92; H, 9.08; N, 7.42. Found: C, 50.86; H, 9.04; N, 7.34. IR, ν/cm^−1^: 3316 w, 3234 w, 3160 w, 2953 m, 2919 m, 2879 w, 2856 m, 1590 m, 1539 vs, 1480 s, 1459 m, 1410 s, 1402 s, 1358 s, 1263 w, 1218 s, 1134 m, 1083 w, 1058 w, 1014 w, 982 m, 940 w, 895 s, 807 m, 792 m, 722 w, 604 s, 535 m, 478 w.

### 3.3. Cooling Sample ***1***

Sample **1** (50 mg) were placed in a glass ampoule connected to the vacuum system using a rubber tube, was kept for 5 h in cold bath (173–183 K range, the mixture of liquid nitrogen and methanol; 77 K, liquid nitrogen in a dewar), was taken out and kept at RT to performed powder diffraction.

### 3.4. PXRD

The XRPD data at room temperature were collected using a Bruker D8 Advance diffractometer (CuK_α_, λ = 1.54 Å, Ni-filter, LYNXEYE detector, geometry reflection). Low temperature experiments were performed at X-ray structural analysis beamline (XSA) of Kurchatov Synchrotron Radiation Source [71]. Monochromatic radiation with a wavelength of 0.74 Å (photon energy 15,498 eV) was used. The sample was placed in a cryoloop of 300 μm in size and rotated around the horizontal axis during the measurement, which made it possible to average the diffraction patterns according to the orientations of the sample. The diffraction patterns were collected by the 2D Rayonix SX165 detector, which was located at a distance of 150 mm with a 29.5° tilt angle, Debye-Scherrer (transmissional) geometry was used with a 400 μm beam size. The exposure time was 5 min. The two-dimensional diffraction patterns obtained on the detector were further integrated to the standard form of the dependence of the intensity on the scattering angle I (2θ) using Dionis software [72]. The cooling rate was 1–1.4 K/min and 2–2.9 K/min at a measurement step of 10 and 40 K, respectively. The heating rate was 4.5 K/min and 2.9–3.2 K/min at a measurement step of 10 and 40 K, respectively.

The diffraction pattern was described by the Rietveld method in the TOPAS software package. The indexing of the sample at 260 K after cooling was performed used the SVD-Index, as implemented in the TOPAS. Then, the powder pattern was refined using the Pawley method to describe δ-**1** lines. The obtained intensity ratio of the peaks in the *hkl*-phase was further fixed and used in modelling of the patterns at other temperatures, with refined scale factor. To obtain the δ-**1** fraction, we compared the obtained scale factors of δ-**1** and α-**1** at 260 and 300 K. Then, assuming that the sample at these temperatures contained only δ-**1** and α-**1,** based on the values of scale factors, we calculated the ratio of the structural factors and used it in the other modeling.

### 3.5. Single Crystal X-ray

The X-ray diffraction data for the crystals α-**1**, β-**1** and γ-**1** were collected on a Bruker D8 Venture diffractometer equipped with a CCD detector (Mo-Kα, λ = 0.71073 Å, graphite monochromator). Semi-empirical absorption correction was applied by the SADABS program [73]. The structures were solved by direct methods and refined by the full-matrix least squares in the anisotropic approximation for non-hydrogen atoms. The calculations were carried out by the SHELX-2014 program package [74] using Olex2 1.2 [75]. The crystallographic parameters for the investigated crystals and the structure refinement details are given in Table 1. The crystallographic data for the structures reported in this paper have been deposited with the Cambridge Crystallographic Data Center (2190847-2190849).

The experimental data at 150 K were obtained as a result of a multi-temperature experiment, in which a phase transition was observed with a change in the crystal system from a monocline *P*2/*n* to a triclinic *P-*1, as a result of which the crystal was gradually destroyed, including during data collection at 150 K. Thus, the obtained experimental data are not ideal, but more than sufficient for an unambiguous determination of the crystal structure.

### 3.6. Quantum Chemical Calculation

The Ab initio calculations were performed using the ORCA 5.0.1 computational package [55]. The calculations of the single-ion anisotropy parameters were based on the state-averaged complete-active-space self-consistent field (SA-CASSCF) wave functions [76,77], complemented by the N-electron valence second-order perturbation theory (NEVPT2) [78,79,80]. The active space of the CASSCF calculations was composed of seven electrons in five d-orbitals of Co^2+^ ions (*S* = 3/2): CAS(7,5). The state-averaged approach was used, in which all 10 quartet (*S* = 3/2) and 40 doublets (*S* = 1/2) states were averaged with equal weights. Both the zero-field splitting parameter (D) and transverse anisotropy (E), based on the dominant spin-orbit coupling contributions from the excited states, were calculated through the quasi-degenerate perturbation theory (QDPT) [81], in which an approximation to the Breit-Pauli form of the spin–orbit coupling operator (SOMF) [82] and the effective Hamiltonian approach [83] was applied. The polarized triple-ӡ-quality basis set def2-TZVP was used for all atoms [84,85]. An auxiliary def2/JK Coulomb fitting basis set was used during the calculation [86]. The splitting of the *d*-orbitals was analysed within the ab initio ligand field theory (AILFT) [87,88]. All ab initio calculations were performed with the geometry of the experimentally determined X-ray structures. The AILFT d-orbitals are described as linear combinations of the *d*-orbitals. Splitting is presented based on the largest coefficients for each orbital.

## Data Availability

The structure parameters of obtained compounds were deposited with the Cambridge Structural Database (CCDC Nos. 2190847 (α-**1**), 2190848 (β-**1**) and 2190849 (γ-**1**); deposit@ccdc.cam.ac.uk or http://www.ccdc.cam.ac.uk/data_request/cif, accessed on 27 November 2022.

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
