# Peer review of "Change in the Electronic Structure of the Cobalt(II) Ion in a One-Dimensional Polymer with Flexible Linkers Induced by a Structural Phase Transition†"

_ijms, 2022, doi:10.3390/ijms24010215_

Round 1

Reviewer 1 Report

This manuscript reports interesting experimental results on three crystal phases of a Cobalt (II) ion in a 2 one-dimensional polymer with flexible linkers. The work is well presented: the several techniques used to characterize and compare the products are appropriate. In this consideration I judge the paper suitable for the publication on International Journal of Molecular Sciences AFTER MINOR REVISION. I suggest the Authors to consider the following points before the submission of the final revised version:

- in the abstract at the lines 25 and 27 twice STP appears instead of SPT

- at page 4, line 143 XRF is instead of XRD

- in the caption of Figure 2, page 4, at line 151 is repeated ‘at the top’

- The XRD patterns of Figures S2 and S3 are collected at T=amb?

- the compositional data of the three crystal phases at all the investigated temperature are reported inside the text. Nevertheless, I suggest the Authors also put Table S1 in the main text because it will be more easily for the reader to follow the description. I suppose the error in the relative amount is at least 1%; in any case it’s better to use the same percentages in the text and in the Table. Several discrepancies are present.

- I suggest the Authors to comment a little more the differences between alpha and beta forms. There is a real difference or simply the crystal studied in beta-1 is more disordered and crystal alpha-1 is more perfect and ordered? Also XRD powder patterns are extremely similar. About peace and war, I ask you not be neutral, but all we must work in order the peace is successful. In other words, we are in the presence of a unique phase, but with two crystals of different quality?

- at page 12, line 395 is recalled Figure S18…but the last one is S16. What is the correct number?

Author Response

- in the abstract at the lines 25 and 27 twice STP appears instead of SPT

Thank you, your comment has been taken into account.

- at page 4, line 143 XRF is instead of XRD

Thank you, your comment has been taken into account.

- in the caption of Figure 2, page 4, at line 151 is repeated ‘at the top’

Thank you, your comment has been taken into account.

- The XRD patterns of Figures S2 and S3 are collected at T=amb?

Yes, at room temperature, the corresponding clarification is given in the captions to the figures

- the compositional data of the three crystal phases at all the investigated temperature are reported inside the text. Nevertheless, I suggest the Authors also put Table S1 in the main text because it will be more easily for the reader to follow the description. I suppose the error in the relative amount is at least 1%; in any case it’s better to use the same percentages in the text and in the Table. Several discrepancies are present.

Thank you, the table S1 was moved in the main text. There is a decrease in the concentration of the content of the δ-1 phase upon cooling to 100 K and subsequent heating to 140 K, but these values are in the same region within the error.

- I suggest the Authors to comment a little more the differences between alpha and beta forms. There is a real difference or simply the crystal studied in beta-1 is more disordered and crystal alpha-1 is more perfect and ordered? Also XRD powder patterns are extremely similar. About peace and war, I ask you not be neutral, but all we must work in order the peace is successful. In other words, we are in the presence of a unique phase, but with two crystals of different quality?

Thank you for the comment. Two phases α-1 and β-1 are quite similar, but they are individual regardless of the quality of the crystals. We tried to solve both X-ray diffraction datasets in monoclinic P and C groups, but it is not possible to convert one to the other and vice versa. Perhaps there is a polymorphism due to statistical disorder. Anyway, these differences allowed us to determine the admixture of the β-1 phase in the sample as a result of the difference in PXRD (see additional file).

Interestingly, these differences are also expressed in the electronic structure. Quantum chemical calculations, in particular, the splitting of initial states and the position of the Kramers doublets (KD), made it possible to determine the pseudo tetrahedral geometry of the crystal field of the Co2+ ion in β-1. Also, HOMO shows that there is no contribution from two oxygen atoms of both chelate carboxylate groups to the common MO. So, we want to study β-1 phase, but do not yet know how to get a pure phase. We do not provide this material in the article, so as not to overload the semantic part of the article, since the sample corresponds to the main α-1 phase.

We empathize with the current situation, unfortunately, some issues of this on this planet are solved collectively, and the team of our group is not enough to change the situation.

- at page 12, line 395 is recalled Figure S18…but the last one is S16. What is the correct number?

Thank you, it was Figure S16, this is the result of changes in a previous text.

Reviewer 2 Report

In this manuscript, the author studied the how structural phase transition can influence the electronic and magnetic properties of Co ions in a 1D polymer. This is a rather comprehensive study with interesting results. Before recommending publication, I have the following concerns/questions:

1.     In Eq. (1), the orbital angular moment L is present, while in Eq. (2), L is not present. Can the author explain why? Is this because the low temperature phase is a very high symmetry phase so L=0?

2.     Following the previous question, there is no zero-field splitting term for spin S in Eq. (1). Why?

3.     For the AC magnetic data, I expect there will be some features (e.g., resonances) when one uses a higher frequency (to GHz). However, here only very low frequencies (<0.1 MHz) are studied. Have the authors tried higher frequencies?

Author Response

  1. In Eq. (1), the orbital angular moment L is present, while in Eq. (2), L is not present. Can the author explain why? Is this because the low temperature phase is a very high symmetry phase so L=0?

Co2+ ions in α-1 are in the pseudo-octahedral coordination environment. When the symmetry of the coordination field is lowered to Oh, the splitting of the 4F and 4P terms are isomorphic. This interaction causes a small admixture between the 4T1(P) level and the ground 4T1(F) level, which is due to the ligand field. Such a mixture provides an additional source of orbital reduction of the low-lying triplet. So, when describing the magnetic properties, it is necessary to take into account the contribution of the orbital momentum to the total magnetic moment. The spin component in eq. 1 is represented by the constants SOC (ζ = Ë—3λ), the commutator , and in the Zeeman operator. The axial and rhombic components are represented by Δax and Δrh. Only in the case of a positive Δax it is possible to describe the system in terms of ZFS with large D ~60-120 cm-1.

For the tetrahedral coordination environment of cobalt ions in γ-1, the Griffith Hamiltonian approach is inapplicable because the splitting of the 4F and 4P terms is not isomorphic. In the ground orbital singlet states of Co2+ ions, the first order orbital angular momentum is quenched.

  1. Following the previous question, there is no zero-field splitting term for spin S in Eq. (1). Why?

The ZFS Hamiltonian considers a purely spin system S=3/2 with no orbital momentum or when it is completely quenched, by definition there is no L there, just as there is no ZFS in GH, because interactions of a different nature are taken into account there.

  1. For the AC magnetic data, I expect there will be some features (e.g., resonances) when one uses a higher frequency (to GHz). However, here only very low frequencies (<0.1 MHz) are studied. Have the authors tried higher frequencies?

AC magnetic susceptibility data were collected on commercially available PPMS-9 magnetometer, which is widely used for such type of measurements. This equipment allows to do AC measurements in a frequency range from 10 to 10,000 Hz. The specified frequency range is sufficient to fully characterize the dynamics of magnetic relaxation. The GHz frequency AC susceptibility measurements are not available not only for our scientific group, but we don't know even one example of such type measurements in the currently available literature.

Round 2

Reviewer 2 Report

The authors addressed my comments satisfactorily. I suggest publication.